# TRPV1 Responses in the Cerebellum Lobules VI, VII, VIII Using Electroacupuncture Treatment for Chronic Pain and Depression Comorbidity in a Murine Model

**DOI:** 10.3390/ijms22095028

**Published:** 2021-05-10

**Authors:** Bernice Lottering, Yi-Wen Lin

**Affiliations:** 1Graduate Institute of Acupuncture Science, College of Chinese Medicine, China Medical University, Taichung 40402, Taiwan; lotteringbernice@gmail.com; 2Chinese Medicine Research Center, China Medical University, Taichung 40402, Taiwan

**Keywords:** chronic pain, depression, cerebellum, TRPV1, electroacupuncture, ST36

## Abstract

Depression is a prominent complex psychiatric disorder, usually complicated through expression of comorbid conditions, with chronic pain being among the most prevalent. This comorbidity is consistently associated with a poor prognosis and has been shown to negatively impact patient outcomes. With a global rise in this condition presenting itself, the importance of discovering long-term, effective, and affordable treatments is crucial. Electroacupuncture has demonstrated renowned success in its use for the treatment of pain and is a widely recognized therapy in clinical practice for the treatment of various psychosomatic disorders, most notably depression. Our study aimed to investigate the effects and mechanisms of Acid-Saline (AS) inducing states of chronic pain and depression comorbidity in the cerebellum, using the ST36 acupoint as the therapeutic intervention. Furthermore, the role of TRPV1 was relatedly explored through the use of TRPV1^−/−^ mice (KO). The results indicated significant differences in the four behavioral tests used to characterize pain and depression states in mice. The AS and AS + SHAM group showed significant differences when compared to the Control and AS + EA groups in the von Frey and Hargreaves’s tests, as well as the Open-Field and Forced Swimming tests. This evidence was further substantiated in the protein levels observed in immunoblotting, with significant differences between the AS and AS + SHAM groups when compared to the AS + EA and AS + KO groups being identified. In addition, immunofluorescence visibly served to corroborate the quantitative outcomes. Conclusively these findings suggest that AS-induced chronic pain and depression comorbidity elicits changes in the cerebellum lobules VI, VII, VIII, which are ameliorated through the use of EA at ST36 via its action on TRPV1 and related molecular pathways. The action of TRPV1 is not singular in CPDC, which would suggest other potential targets such as acid-sensing ion channel subtype 3 (ASIC3) or voltage-gated sodium channels (Navs) that could be explored in future studies.

## 1. Introduction

Neuropsychiatric disorders display a complex relationship with the physical manifestations of disease in patients, often presenting with corresponding clinical signs and common pathological pathways [1]. As a leading cause of global disability [2], chronic pain and depression comorbidity (CPDC) has garnered the attention of researchers worldwide and significantly contributed to a vast array of negative socio-economic effects affecting individuals and communities alike [3]. Understanding and identifying the underlying molecular processes involved in CPDC provides potential therapeutic targets to address the clinical manifestations thereof. To this end, the similarity of symptomatic presentations between chronic pain and depression underline shared biological mechanisms, such as neuroinflammation and central sensitization [4]. Accordingly, chronic pain states can be observed in a multitude of inflammatory conditions [5], and the associations of neurobiological conditions pertaining to the development of depression also appear to be triggered or reinforced by these inflammatory processes [6].

Transient receptor potential vanilloid 1 (TRPV1) is a widely accepted marker of inflammation observed in various nociceptive responses [7]. As the first of six members of the TRPV subfamily [8], its function as a nonselective calcium-permeable cation channel is observed through the regulation of both neuronal and non-neuronal activities [9]. Specifically, the neurobiological correlates of CPDC can be observed through widespread expression of TRPV1 [10], with a resultant pathological cascade implicating signaling molecules that are well-established in neuronal plasticity, central sensitization, neuroinflammation, nociception, and various cognitive aptitudes.

Despite the evident existence of various pathological mechanisms underlying the manifestation of CPDC, current therapeutic interventions are limited with regard to treating this specific comorbidity comprehensively. The use of analgesics or antidepressants aims to specifically attenuate the symptoms of either pain or depression, but not necessarily both. Therefore, identifying targets that treat CPDC as a singular disease, as opposed to individual symptoms, would allow for more reliable and beneficial treatments. Furthermore, the large number of reported side effects and risks associated with the use of analgesics and antidepressants highlights the need for safer treatment regimens [11]. Traditional pain medications, such as opioids and non-steroidal anti-inflammatory drugs (NSAIDs), are indeed effective in addressing the nociceptive response, but they are limited by adverse effects, tolerance, and potential for addiction [12]. Likewise, the clinical adverse effects of anti-depressive drugs such as selective serotonin reuptake inhibitors (SSRIs), norepinephrine reuptake inhibitors (NRIs), or tricyclic antidepressants (TCAs) can decrease the compliance and delay recovery in patients [13]. The prevalence of suicidality, emotional blunting, and withdrawal effects are the most prominent of symptoms experienced by patients, alongside simple non-response, and the justification of the use of these treatments is often controversial with respect to the risk-to-benefit ratio [14].

The use of acupuncture has been widely accepted in the treatment of various diseases and is considered superior in its efficacy with regard to pain management [15]. Scientific evidence further corroborates the value of acupuncture in neuropsychiatry treatment [16], with its efficacy in depression being extensively investigated and scientifically established [17]. The commonality of chronic pain and depression has been explored through shared biological pathways and neurotransmitters [18], with acupuncture treatment activating the endogenous pain control systems whilst simultaneously having beneficial effects on emotional, psychological, and interpersonal domains [19]. The modern clinical application of acupuncture involves the use of an electrical stimulator to produce increased signal and response mechanisms within the central nervous system [20]. The application of this stimulating method, known as electroacupuncture (EA), has displayed a proficient aptitude in the treatment of a large variety of chronic pain conditions [21,22], as well as several psychological conditions [16]. Furthermore, the neurological signaling cascades involved in the mediation of CPDC can be activated through ST36 (zusanli), a TRPV1 responding mechanosensitive acupoint generally used in clinical regimes [23].

The mediating role of acupuncture in CPDC has garnered the attention of researchers worldwide, to which end specific biochemical pathways within the central and peripheral nervous systems have been investigated. Changes within the cerebellum, a brain region classically involved in motor processing and cognitive function, have been rudimentarily implicated in this diseased state [24]. The conventional model of cerebellar function and organization ordains involvement in motor control through coordination of movement and learning, but it has also been directly implicated in the effective processing of pain [25]. Recent literature has discovered that accelerated brain ageing accompanies chronic pain conditions, whereby lower cerebellar gray matter has been identified as a key pathological contributor [26]. Moreover, the aberrant neural functional activity of chronic pain conditions involves the activation of a multitude of brain areas, with cerebellar involvement being consistent with network changes associated with central sensitization [27]. Additionally, modernized findings explore further roles of the cerebellum in neuropsychiatric disorders [28] and serve to define a multitude of important physiological responses that require additional attention from researchers in the field of nociception and psychiatry. Non-invasive stimulation of the cerebellum provides a potential therapeutic approach in the treatment of cognitive deficits related to depression [29], and this suggests that cerebellar involvement in depression and pain responses is crucial. Whilst an increased interest in cerebellar involvement in both chronic pain and depression has been observed, potential cerebellar contributions to pathological mechanisms attendant to CPDC have rarely been explored and should indeed be investigated.

On the basis of the aforementioned work, this study aimed at clarifying the efficacy of EA treatment and characterizing the modification of various molecular substrates in the cerebellum in CPDC disorder. An Acid-Saline (AS) injection model, a frequently used model in animal model laboratory research that produces chronic states of hyperalgesia [30], was used to induce states of CPDC. We hypothesized that AS injections would cause alterations in the cerebellum as a result of the CPDC, which would be attenuated through the use of EA at ST36 via actions on TRPV1 and associated downstream molecules.

## 2. Results

### 2.1. The Effect of EA Treatment on Mechanical and Thermal Nociceptive Behavioral Responses

Electroacupuncture significantly attenuated mechanical and thermal hyperalgesia in the chronic pain and depression comorbidity model. As shown in Figure 1A,B, mechanical and thermal nociceptive responses did not differ among the five groups under basal conditions. The Control group represented normal values, as no Acid-Saline injections resulted in pain sensations. According to the von Frey testing results (Figure 1A), mechanical sensitivity indicated a significantly higher pain threshold observed across the 4 CPDC-induced groups, namely AS, AS + EA, AS + SHAM, and AS + KO, before treatment intervention, from day 3 to day 14, indicating a successful induction of chronic pain behavior. A similar tendency was observed in the results of thermal sensitivity measured in the Hargreaves’ test (Figure 1B), with the AS, AS + EA, and AS + SHAM groups displaying increased nociceptive responses when compared to the Control group. However, no significant difference was observed when comparing the AS + KO group to the Control group, suggesting that the thermal sensitivity of TRPV1 KO mice was accurately reduced according to the role of TRPV1 in thermal hyperalgesia.

Day 21 depicted the first significant difference observed in the AS + EA group after receiving three EA treatments at ST36 (2 Hz/20 min). The von Frey filament outcomes were significantly increased in the AS + EA group when compared to the AS, AS + SHAM, and AS + KO groups, whilst the difference between the AS + EA and Control group was also still statistically significant. Moreover, the paw withdrawal latency of the Hargreaves’ thermal sensitivity response was proportionately decreased in the AS + EA group when compared to the no treatment and sham treatment groups of AS and AS + SHAM, respectively. Again, the AS + KO group remained unchanged, and the AS + EA group was still significantly different when compared to the Control group, but displayed a tendency for reduction in nociceptive responses. Day 28 revealed the true efficacy of EA treatment at ST36 (2 Hz/20 min), with clear significant differences observed when comparing the AS + EA group to the AS and AS + SHAM groups in both the mechanical and thermal sensitivity test results. EA significantly reduced mechanical and thermal hyperalgesia, as continuous reductions in both examinations were observed from day 21 to day 28. However, the difference between AS + EA and Control groups was still statistically significant on day 28, although the tendency of efficacy indicated beneficial results, suggesting that prolonged treatment of EA could potentially completely attenuate chronic pain.

### 2.2. The Effect of EA Treatment on Depression-Like Behavior in Open Field Testing (OFT) and Forced Swimming Test (FST) Behavioral Responses

Electroacupuncture significantly attenuated depressive behavior in the chronic pain and depression comorbidity model as observed in the OFT and FST behavioral responses. As shown in Figure 1C, there was no significant difference observed in the total distance of the OFT on day 28, which was comparatively similar and indicated no physical impediments among the five groups. The distance in the center zone (Zone 3) was significantly decreased in the AS, AS + SHAM, and AS + KO groups when compared to the Control group. However, the AS + EA group indicated no significant difference when compared to the Control group, depicting attenuation of depressive-like behavior resulting from EA treatment at ST36 (2 Hz/20 min) (Figure 1D). This result was supported by similar tendencies as observed in the FST on day 28 as shown in Figure 1E,F. The immobility duration was significantly increased in the AS, AS + SHAM, and AS + KO groups, which is considered a standard criterion for the expression and recording of depression. Again, EA significantly attenuated this behavior through a substantial reduction in the immobility duration, whereby the associated AS + EA group depicted no difference when compared to the Control group, thus indicating an amelioration of depressive-like behavior through the applied treatment.

### 2.3. The Effect of EA Treatment on CPDC Associated Pathways and Related Molecules in the Cerebellum Lobules VI, VII, and VIII

After the initial confirmation of successful CPDC induction based on the relevant behavior test results, the brain samples were collected in order to detect the related protein modification within the cerebellar lobules VI, VII, and VIII. The effect of CPDC on TRPV1 and related molecules was investigated, and the associated molecular foundations of EA were further explored. The Control was considered and ideal value and standardized to 100%. Numerical results were displayed as a standard error of mean (S.E.M), and a *p* value of 0.05 was considered statistically significant (Table 1).

First the cerebellum lobule VI (Figure 2) was investigated. It was found that the classic nociceptive receptor of (A) TRPV1 displayed significant decreases in the AS and AS + SHAM groups, which received the AS injections without any true treatment intervention in comparison to the standard of the Control group. These significant decreases were augmented in the AS + EA group, which underwent EA treatment. As anticipated, the AS + KO group displayed an absence of TRPV1 as a result of TRPV1 gene deletion. Accordingly, the involvement of (B) pmTOR, (E) pPI3K, (H) pAkt, and (N) pERK, which are the principal pathways of MAPK, was investigated. Furthermore, important markers of (G) pPKCε and (L) pPKAIIα were examined for their role in the induction and maintenance of hyper-nociception. The roles of (J) pNFκB and (M) pCREB as molecular mediators within the nucleus were also explored. Lastly, the data of (F) NMDAR1, (I) TrkB, and (K) GABAAα1 were collected and researched for the relevancy with regard to depression pathologies. Across all of these biomarkers, homologous tendencies in the protein densities were observed, whereby significant decreases were discerned in the cerebellum lobule VI of AS mice when compared to the Control group. This phenomenon was correspondingly observed in the AS + SHAM group, which indicated a similarity in diminished expression tendency when compared to the Control group. In contrast, the apparent decrease in both the AS and AS + SHAM groups was significantly reversed through the treatment of EA as well as TRPV1 gene deletion. Both the AS and AS + KO groups displayed significant differences when compared to the AS and AS + SHAM groups, but none when compared to the Control group. Conversely, the voltage-gated sodium (C) Nav1.7 and (D) Nav1.8 receptors displayed a significantly increased expression within the AS and AS + SHAM groups in comparison to the Control group. In this respect, the AS + EA and AS + KO groups were significantly diminished when compared to the AS and AS + SHAM group and depicted a near normalized state.

Next, the cerebellum lobule VII (Figure 3) was concurrently dissected and examined. Accordingly, a similar tendency was observed in a majority of the aforementioned molecules. Specifically, the data indicated a statistically significant underexpression in (A) TRPV1, (B) pmTOR, (E) pPI3K, (F) NMDAR1, (G) pPKCε, (H) pAkt, (I) TrkB, (J) pNFκB, (K) GABAAα1, (L) pPKAIIα, (M) pCREB, and (N) pERK protein levels of the AS and AS + SHAM groups. This evident underexpression was ameliorated in both the AS + EA and AS + KO groups, which displayed no statistical variance when compared to the Control group, therefore depicting states of increased expression when observed alongside the under-expression of the AS and AS + SHAM groups. Interestingly, the (C) Nav1.7 and (D) Nav1.8 receptors demonstrated no significant variances among the five subject groups, regardless of wild-type or knockout mice, suggesting location-specific involvement of these two receptors.

In addition, we analyzed the involvement of the aforementioned 14 proteins in the cerebellum lobule VIII among the four wild-type groups and the knockout group (Figure 4). The AS injection induced significant depressions in AS and AS + SHAM groups when compared to the normal Control group in 12 of these proteins, namely (A) TRPV1, (B) pmTOR, (E) pPI3K, (F) NMDAR1, (G) pPKCε, (H) pAkt, (I) TrkB, (J) pNFκB, (K) GABAAα1, (L) pPKAIIα, (M) pCREB, and (N) pERK levels. A significant increase was observed in the CDPC incitement AS + EA group that underwent EA treatment, as well as the AS + KO group, which is missing the TRPV1 receptor. Again, the results depicted a divergent trend of statistical intensification in the AS and AS + SHAM groups of the voltage-gated sodium (C) Nav1.7 and (D) Nav1.8 receptor activity, whereby an apparent overexpression was conversely observed in AS and AS + SHAM groups, that was for a second time decreased through the treatment of EA or TRPV1 gene deletion as observed in the AS + EA and AS + KO groups. This tendency was identically observed in cerebellum lobule VI, but not in VII. These results allude to the involvement of TRPV1 and related molecules in the acquisition and expression of CPDC within the cerebellum of mice, and they further provide novel mechanisms of EA-dependent molecular function and action.

### 2.4. The Effect of EA Treatment on Protein Expression Alteration in Cerebellum Lobules VI, VII, and VIII as Observed by Immunofluorescence

Lastly, the quantitative evidence of immunoblotting in cerebellum lobules VI, VII, and VIII was further observed through the use of immunofluorescence depicted in Figure 5 and Figure 6. The images of the cerebellum lobules VIa, VII, and VIII displayed analogous tendencies of reduced expression of TRPV1 and pNFκB protein staining in the AS and AS + SHAM groups, in contrast to the Control group. Additionally, the AS + EA and AS + KO groups were visibly augmented and accordingly presented significant increases in protein density. However, since cerebellar lobule VI was observed in two different areas as a result of the large zone, and respectively divided into VIa and VIb (Figure 5), a different trend was observed in pNFkB expression. Specifically, a converse tendency of insignificant variance was observed in cerebellar lobule VIb (Figure 5B) of the pNFkB protein levels across all five groups. Similar results were also observed in cerebellum lobules VII and VIII (Figure 6).

## 3. Discussion

Chronic pain and depression are considered to be extremely detrimental, long-lasting, disabling conditions that significantly reduce the quality of life, and when presenting as comorbidity, an observable increase in the number of individuals presenting with this disorder underlines the urgent need to improve the management and treatment of these patients [31]. Evidence suggests that current treatment interventions, and their related side effects, are potential contributors to compounded expressions of CPDC, in that the severity of pain and depression is not always diminished but conversely drives increased expressions of pain and depressive symptoms [32]. Acupuncture treatment offers numerous advantages over traditional analgesic and antidepressant options, as evidence supports observable absences of the detrimental side effects associated with these conventional intervention approaches, such as addiction, psychotic events, gastrointestinal damage, and more [33]. Our investigations were aimed at identifying the role of various neuromodulators in CPDC and the efficacy of EA in the treatment thereof. To this end, the involvement of TRPV1, a calcium ion channel, has been largely evidenced as a functional component of neuromodulation expressed in a wide array of pain conditions [34,35], and it has further been targeted for its behavior-regulating capacity via various biological systems [36]. In view of the diverse array of molecular aspects relating to the comorbid relationship of CPDC, the cerebellum was targeted for its renowned participation in both modulating states of emotional processing and pain responses [37,38].

This study observed pain-like behaviors in accordance with the widely used scientifically recognized von Frey and Hargreaves testing methods [39]. Furthermore, depressive behaviors were identified in concordance with the commonly accepted open-field and forced-swim testing methods [40]. An increase in mechanical and thermal hyperalgesia confirmed that AS injections successfully evoked pathological nociceptive responses. Furthermore, the decreased distance travelled in the center zone of the OFT, alongside increased immobility recorded in the FST on day 28, demonstrated a successful induction of depression. Broadly, our results validated the induction of CPDC, and the significant difference between the AS + EA and AS groups from days 14 to 28, signifying the efficacy of EA at ST36 in reducing chronic pain and depression. Interestingly, this difference displayed a gradual therapeutic tendency, which confirms that continued or prolonged use of EA results in sustained prophylactic effects [41]. The lasting effects of EA specificity at ST36 were distinctly identified through amelioration of CPDC not reflected in the SHAM group, and this is consistent with evidence supporting that the beneficial effects of acupuncture occur earlier, are larger, and last longer [42] than SHAM acupuncture, with the needling of specific acupoint locations being essential contributors to the observed treatment benefit [43]. As hypothesized, our results confirm and strengthen previous key findings that acupoint specific treatments have clinically relevant effects that appropriately persist over a long-term period. In addition, the TRPV1 KO group, which received no treatment, displayed states of sustained hyperalgesia and maintained depressive-like behavior, which would suggest that the role of TRPV1 in CPDC is not singular [44].

The well-known reciprocal relationship between pain and depression has been extensively observed through widespread preclinical research, with a multitude of molecular aspects that link these pathologies being identified. In this regard, not only TRPV1 but also a variety of related key neuromodulators has been shown to play an important role in this comorbidity [45]. Accordingly, our findings displayed significantly decreased protein densities in the cerebellum lobules VI, VII, and VIII of the AS and AS + SHAM groups when compared to the Control group, which provides novel evidence of CPDC mechanisms in the cerebellum. As a largely relevant nociceptor implicated in both pain and mood disorders, TRPV1 plays an essential role in the expression of CPDC. The distribution and abundance of TRPV1 across both exogenous and endogenous systems has been clearly described [46], with a decrease in expression being associated with concurrent downregulation of affiliated molecular interaction, observed in CPDC [10]. Specifically, our findings correlate evidence that TRPV1 expression in the cerebellum correlates with observable functional changes of associated structures [47].

GABAAα1 signaling mechanisms have been revealed by structural pharmacology [48] and have been evidenced for their large operative role in both pain and depression pathologies [49]. Evidence suggests that early-stage neuroinflammation, potentially caused by acute injury, inhibits GABAAα1 expression through astrocyte activation, which subsequently downregulates the BDNF-TrkB signal pathway and results in an impairment of neurogenesis, thus affecting depression-like symptoms concurrently associated with chronic pain comorbidities [50]. Accordingly, TrkB receptor modulation has been identified for its apparent role in nociceptive signaling [51], with further anti-depressive effects being linked to BDNF-TrkB-ERK/Akt signaling activation and upregulation [52]. Additionally, the phosphorylation of certain TrkB residues is largely responsible for inducing intracellular Ca^2+^ release and activation of calcium-dependent signaling, which represents the primary regulator of synaptic plasticity [53], and accordingly can be linked to TRPV1 functioning as a Ca^2+^ channel respondent. Hence, the observable downregulation of both the TrkB and TRPV1 receptors in our results can be evidenced as direct contributors to CPDC pathologies. The abnormalities of these aforementioned receptors are further extended to impairment of function of the NMDAR1 receptors in the case of CPDC, which is observed at the cellular and neurochemical levels [54]. Pathological states associated with neurodegeneration and dysfunction have been associated with decreased expression of NMDAR1, and an increase in receptor activity being associated with neuroprotective benefits [55]. This was directly observed in our results, whereby the AS + EA group attenuated the significant decrease in protein density observed in the AS and AS + SHAM group.

Localized and integrated neural networks of the aforementioned receptors share in the further cascade signaling of corresponding receptors and mediators, most notably the intercellular kinase pathway, protein kinase C, PI3K/Akt pathway, and the mitogen-activated protein kinase (MAPK) pathways [9,56,57]. To this end, a pathological downregulation of these molecular constituents, namely PKCε, PKAIIα, ERK, PI3K, Akt, and mTOR, has been concurrently observed in various chronic pain and depressive states, and this provides evidence as to substantial interactions with the TRPV1 receptor, with an affinity for their involvement being observed in changes induced as a result of acupuncture therapies [58]. Among the transcription factors affiliated with CPDC, the cyclic AMP response element binding (CREB) protein displays a modulating role in the comorbidity of chronic pain and depression. Reduced expression of CREB has been associated with pain-induced depressive behaviors [59], and a reversal of these phenomena through therapeutic interventions has corroborated our findings that increased CREB expression has antidepressant benefits, which in turn increase tolerability to pain [60].

The converse over-expression of the voltage-gated sodium channels of Nav1.7 and Nav1.8 in the cerebellar lobules VI and VIII, but not in VII, suggest locational specificity, and this is similarly observed in acute pain states within the cerebellum [61]. Loss of function of the Nav1.7 leads to allodynia without other neurodevelopmental alterations such as depression, whereby repression of Nav1.7 has been associated with long-lasting analgesia [62]. Moreover, increased states of psychological stress result in increased expression of Nav1.7 and potentiate the pathogenesis of chronic pain [63]. In addition, Nav1.8 is associated with signaling in nociceptive sensory afferents, and hyper function of this receptor is coordinated with hyperalgesia [64]. Both of these receptors have displayed involvement in nociceptive responses associated with TRPV1 involvement, and they concurrently display positive tendencies in the treatment of EA for conditions of chronic pain and depression [65,66].

The abundant expression and functional participation of TRPV1 at ST36 is elicited through effective EA treatment [23] and proposes possible neurological signaling involvement in the amelioration of various disorders through its well-defined analgesic benefits and correctional psychoactive function ability. Accordingly, these findings suggest that CPDC can be improved through increases in TRPV1 signaling, which attenuates this co morbidity in the cerebellar lobules VI, VII, and VIII. Furthermore, EA displays a positive tendency to treat CPDC as identified in the cerebellum via upregulation of TRPV1, associated receptors, and downstream molecules. This study has provided improved understanding of the molecular mechanisms in EA treatment with respect to the affiliated neurological signaling involved in CPDC. Furthermore, novel insights into the subsequent application of TRPV1 as a treatment target for CPDC have been provided. However, limitations of this study are present. Firstly, other brain areas involved in the comorbidity of chronic pain and depression were not included. Additionally, various other neurotransmitters, receptors, and molecules that may also be affected by CPDC were not defined and could present opportunities for future research. Other antagonists of neurotransmitters could also be investigated in future studies in order to better define the mechanisms involved in EA treatment. Lastly, clinical trials could be performed in order to provide clarity for the application of TRPV as a treatment target for CPDC.

In conclusion, we determined significant differences in the pain and depression behaviors in AS-induced CPDC mice. We conclude that AS-induced CPDC directly influences the acquisition and expression of TRPV1, pmTOR, Nav1.7, Nav1.8, pPI3K, NMDAR1, pPKCε, pAkt, TrkB, pNFκB, GABAAα1, pPKAIIα, pCREB, and pERK in the cerebellum via either Western blot or immunofluorescence evidence. Furthermore, 2 Hz EA treatment at ST36, and TRPV1 gene deletion, can regulate the protein modifications observed in CPDC and subsequently regulate the affiliated symptoms of chronic pain and depression via these nociceptors and related downstream molecules in the cerebellum lobules VI, VII, and VIII (Figure 7). TRPV1 is not singular in CPDC, which would suggest other potential targets that could be explored in future studies. Furthermore, we observe that TRPV1 does not necessarily respond to all behaviors. Several receptor/ion channels such as ASIC3 or voltage-gated sodium channels were also reported to be involved in this model.

## 4. Materials and Methods

### 4.1. Experimental Animals

In total, 30 female C57BL/6 mice were used. Twenty-four female C57BL/6 wild-type (WT) mice (BioLASCO, Yilan, Taiwan Co., Ltd., Taiwan) and 6 female C57BL/6 TRPV1 knockout (KO) mice (Jackson Lab, Bar Harbor, ME, USA) aged 8–12 weeks and weighing 18–23 g were used in this study. The animals were maintained in a temperature-controlled room (25 ± 2 °C), at relative humidity of 60 ± 5%, under a 12 h light-dark cycle (from 8.00 a.m. to 8.00 p.m.) with standard food and water provided ad libitum. To minimize the suffering of the animals, at the appropriate point in the experiment, mice were anesthetized using 1–2% isoflurane gas and sacrificed with a lethal overdose of chloral hydrate (400 mg/kg) via intraperitoneal injection. The usage of these animals was approved by the Institute of Animal Care and Use Committee of China Medical University (Permit no. 2017-374), Taiwan, following the Guide for the use of Laboratory Animals (National Academy Press). Mice were subdivided into 5 total groups, with 4 groups being randomly selected: Control group (Con), Acid-Saline group (AS), Acid-Saline with Electroacupuncture group (AS + EA), Acid-Saline with Sham EA (ICS + SHAM), and a final group of non-randomly selected TRPV1 knockout mice exposed to Acid-Saline injections with no treatment intervention applied (AS + KO). Efforts were made to minimize the number of animals used.

### 4.2. AS Model

The chronic pain and depression comorbidity inducing mouse model used in this study was induced with Acid-Saline injections. Briefly, all 30 mice were kept at room temperature, 24 ± 2 °C, with ad libitum feeding of the standard laboratory diet and water before use, prior to testing. In the AS model experiments, utilizing the AS, AS + EA, AS + SHAM, and AS + KO groups, 4 mice were caged together in a plexiglass cage (13 × 18.8 × 29.5 cm) covered with stainless steel mesh for the duration of the experiment. At the start of the experiment (day 0) all mice, except controls, were anaesthetized with 1% isoflurane and received dual injections of 20 µL acid saline into the right gastrocnemius muscle on day 0 and day 1 (acidic saline was prepared in 10 mM 2-[*N*-morpholino]ethanesulfonic acid and further adjusted to pH 4.0 with 1 N NaOH) to model chronic pain.

### 4.3. EA Treatment

Mice were anaesthetized using 1–2% isoflurane gas. Under anesthesia of 1% isoflurane, a pair of stainless steel acupuncture needles (1 inch, 36G, YU KUANG, Taichung, Taiwan) was bilaterally inserted perpendicularly at a depth of 3–4 mm into the murine equivalent of the human ZuSanLi (ST36) acupoints. The murine ST36 is located approximately 4 mm below and 1–2 mm lateral to the midpoint of the knee in mice [67]. A bilateral subcutaneous transverse insertion of acupuncture needles into the scapular region was considered to be the SHAM acupoint. In both the true electroacupuncture (AS + EA) group and the false electroacupuncture group (AS + SHAM), electrical stimuli were delivered by an EA Trio 300 stimulator (Ito, Japan) at an intensity of 2 mA for 20 min at 2 Hz with a pulse width of 150 μs. The electrical stimulation caused slight visible muscle twitching around the area of insertion. The electrical stimulation was applied to the AS + EA and AS + SHAM groups on day 14, 16, 18, 21, 23, and 25 to total 6 treatment sessions (three times a week, every other day from week 2 to week 4).

### 4.4. Behavioral Examination

#### 4.4.1. Nociceptive Behavioral Examination

The mechanical and thermal nociception and sensitivity were each tested 6 times throughout the experiment: firstly on day 0, at the start of the experiment; secondly on day 2, at the end of the AS chronic pain inducing protocol but before the first treatment intervention in the AS + EA and AS + SHAM groups; and then on day 7, 14, 21, and 28, after the final EA treatment but before sacrifice. All mice were transported to the behavior analysis room and were adapted to the environment for 30 min to 1 h before the commencement of the behavior tests. All experiments were performed at room temperature (24 ± 2 °C), and the stimuli were applied only when the animals were calm but not sleeping or grooming.

First, the von Frey assessment was conducted. Mechanical sensitivity was measured by testing the force of responses to stimulation with 6 applications of electronic, calibrated von Frey filament (IITC Life Science Inc., Woodland Hills, CA, USA). Subjects were placed onto a metal mesh (75 × 25 × 45 cm), covered with a plexiglass cage (10 × 6 × 11 cm), and acclimated for 30 min. Subjects were then mechanically stimulated by the tip of the filament at the plantar region of the right or left hind paw. The filament gram counts were recorded when the stimulation caused the subject to withdraw its hind paw. A cut-off pressure of 20 g was set to avoid tissue damage.

Second, the Hargreaves’ assessment was conducted. Thermal pain was measured by testing the time of responses to thermal stimulation with 6 applications using Hargreaves’ test IITC analgesiometer (IITC Life Sciences, SERIES8, Model 390G, Woodland Hills, CA, USA). Subjects were placed in a plexiglass cage on top of a glass sheet and acclimated for 30 min. The thermal stimulator was positioned under the glass sheet, and the focus of the projection bulb was aimed exactly at the middle of the plantar surface of the right or left hind paw. A mirror attached to the stimulator permitted visualization of the plantar surface. A cut-off time of 20 s was set to prevent tissue damage. In the thermal paw withdrawal test, the nociception threshold was assessed using the latency of paw withdrawal upon stimulus and was recorded when the constant applied heat stimulation caused the subject to withdraw its hind paw.

#### 4.4.2. Depression Behavioral Examination

The Open Field Test (OFT) and Forced Swimming Test (FST) were observed 5 times throughout the experiment to measure depressive-like behavior. First on day 0, at the start of the experiment, and then on day 7, 14, 21 and 28 after the final EA treatment, but before sacrifice, using the EthoVision XT 8.5 (Noldus Information Technology, Wageningen, The Netherlands) video-tracking software.

The OFT box was composed of acrylic plastic that formed a 30 × 30 cm square with a wall height of 15 cm. The box was divided into nine equal squares. Each mouse was placed in the center zone at the beginning and allowed to explore the maze for 15 min. The distance of crossing the central zone, the duration in the center area, and the total movement in the open field were analyzed. A low frequency of crossing the central zone or a short duration of time spent within the central zone was considered a validation of depression.

The FST apparatus was a plastic cylinder (47 cm height, 38 cm inside diameter) containing 38 cm of water at 25  ±  1 °C. The water level was deep enough (18 cm) so the tail of the mouse never touched the bottom. The water was replaced between each test. The mice were exposed to forced swimming, for an uninterrupted period of 5 min, and the duration of immobile behavior was measured. After the test, the mice were removed and dried with a towel before being returned to their home cages. Increased immobility in the forced swimming test was indicative of depression-like behavior.

The behavioral examiners were not blinded. The completion of the 5th and final behavior test marked the end of the experiment, and mice were sacrificed via a chloral hydrate overdose (400 mg/kg) via intraperitoneal injection and then cervical dislocation. The cerebellum lobules VI, VII, and VIII were collected for immunoblotting and immunofluorescence analysis.

### 4.5. Immunoblotting

The relevant tissues were initially placed on ice immediately after excision and later stored at −80 °C, pending protein extraction. Total proteins were homogenized in cold radio immunoprecipitation (RIPA) lysis buffer containing 50 mM Tris-HCl pH 7.4, 250 mM NaCl, 1% NP-40, 5 mM EDTA, 50 mM NaF, 1 mM Na_3_VO_4_, 0.02% NaN_3_, and 1× protease inhibitor cocktail (AMRESCO). The extracted proteins were subjected to 8% SDS-Tris glycine gel electrophoresis and transferred to a PVDF membrane. The membrane was blocked with 5% non-fat milk in TBS-T buffer (10 mM Tris pH 7.5, 100 mM NaCl, 0.1% Tween 20) and incubated with a primary antibody in TBS-T with 1% bovine serum albumin (BSA) for 1 h at room temperature with the following antibodies (1:1000, Alomone, Jerusalem, Israel): anti-tubulin, β-actin, anti-TRPV1, anti-pmTOR, anti-Nav1.7, anti-Nav1.8, anti-pPI3K, anti-NMDAR1, anti-pPKCε, anti-phosphorylated (p)-Akt, anti- TrkB, anti-pNFκB, anti-GABAAα1, anti-pPKAIIα, anti-pCREB, or anti-pERK in TBS-T with 1% bovine serum albumin (BSA). Horseradish peroxidase-conjugated AffiniPure goat anti-mouse, goat anti-rabbit, or donkey anti-goat secondary antibodies (1:5000) were incubated with the membranes for 2 h’s incubation at room temperature. The protein bands on the membranes were visualized using an enhanced chemiluminescent substrate kit (Pierce; Thermo Fisher Scientific, Inc. Waltham, MA, USA) with LAS-3000 Fujifilm (Fuji Photo Film Co. Ltd. Tokyo, Japan). The image densities of the specific bands were quantified by National Institutes of Health (NIH) ImageJ software (version 1.8.0). All positive controls had been tested prior to experimentation and are included as a defined standard in the supplier details.

### 4.6. Immunofluorescence

Animals were euthanized via 5% isoflurane inhalation and intracardially perfused with normal saline followed by 4% paraformaldehyde. The brain was immediately excised and post-fixed with 4% paraformaldehyde at 4 °C for 3 consecutive days. The tissues were placed in 30% sucrose for cryoprotection overnight at 4 °C. The brain was embedded in optimal cutting temperature (OCT) compound and rapidly frozen using liquid nitrogen before storing at −80 °C. Frozen segments were cut at 20 μm width on a cryostat then instantaneously placed on glass slides. The samples were fixed with 4% paraformaldehyde, then incubated with blocking solution, consisting of 3% BSA, 0.1% Triton X-100, and 0.02% sodium azide, for 1 h at room temperature. After blocking, the samples were incubated with the primary antibody (1:200, Alomone, Jerusalem, Israel), TRPV1 and pNFκB, prepared in 1% bovine serum albumin solution at 4 °C overnight. The samples were then incubated with the secondary antibody (1:500), 488-conjugated AffiniPure donkey anti-rabbit IgG (H + L), 594-conjugated AffiniPure donkey anti-goat IgG (H + L), and peroxidase-conjugated AffiniPure donkey anti-mouse IgG (H + L) for 2 h at room temperature before being fixed with cover slips for immunofluorescence visualization. The samples were observed by an epi-fluorescent microscope (Olympus, BX-51, Tokyo, Japan) with 20× numerical aperture (NA = 0.4) objective. The images were analyzed by NIH ImageJ software (Bethesda, MD, USA).

### 4.7. Statistical Analysis

Statistical analysis was performed using the SPSS statistic program. All statistical data are presented as the mean ± standard error (SEM). Statistical significance among all groups was tested using the repeated-measures ANOVA test, followed by a post-hoc Tukey’s test. Values of *p* < 0.05 were considered statistically significant.

## Figures and Tables

**Figure 1 ijms-22-05028-f001:**
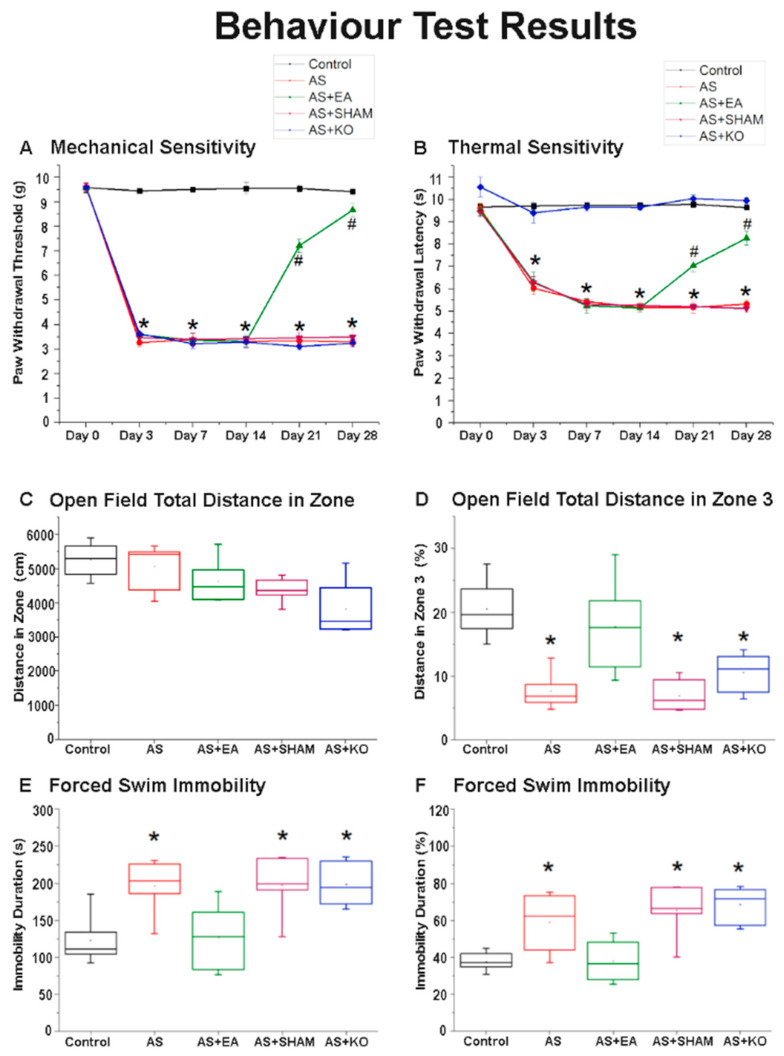
Comparative graph of paw withdrawal threshold and latency of the Acid-Saline (AS)-induced chronic pain subjects after electroacupuncture (EA) treatment and transient receptor vanilloid member 1 (TRPV1) gene deletion (KO). Control, AS, AS + EA, AS + SHAM, and AS + KO were tested according to (**A**) mechanical von Frey and (**B**) thermal Hargreaves’ nociceptive sensitivities. * *p* < 0.05 means when compared with the baseline of control group. # *p* < 0.05 means when compared with the AS and AS + SHAM groups. (**C**,**D**) Open Field testing and (**E**,**F**) Forced Swimming testing to depict depressive-like behavior. * *p* < 0.05 means when compared with the baseline of control group *n* = 6 for five groups.

**Figure 2 ijms-22-05028-f002:**
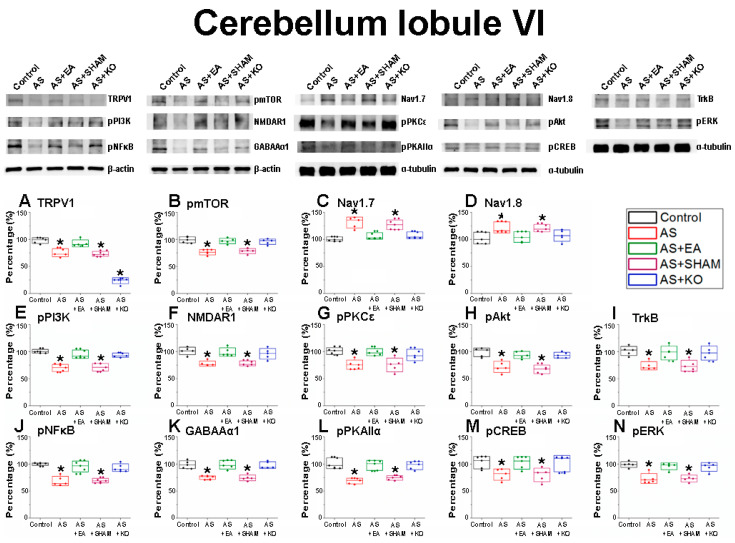
The expression levels of nociceptive receptors and associated molecules in cerebellum lobule VI. The immunoblotting images depict five lanes of protein in the following order: Control, AS, AS + EA, AS + SHAM, and AS + KO groups. There are significant decreases in protein expression in the AS and AS + SHAM groups of (**A**) TRPV1, (**B**) pmTOR, (**E**) pPI3K, (**F**) NMDAR1, (**G**) pPKCε, (**H**) pAkt, (**I**) TrkB, (**J**) pNFκB, (**K**) GABAAα1, (**L**) pPKAIIα, (**M**) pCREB, and (**N**) pERK levels, which were significantly attenuated in the AS + EA and AS + KO groups, depicting no difference when compared to the Control group. Conversely, the protein expressions of the AS and AS + SHAM groups were significantly increased in (**C**) Nav1.7 and (**D**) Nav1.8 when compared to the Control group. Correspondingly, the AS + EA and AS + KO groups were augmented and showed no significant difference in comparison to the Control group. Accordingly, the protein density of the AS + KO group revealed a predicted decrease in (**A**) TRPV1 (* *p* < 0.05).

**Figure 3 ijms-22-05028-f003:**
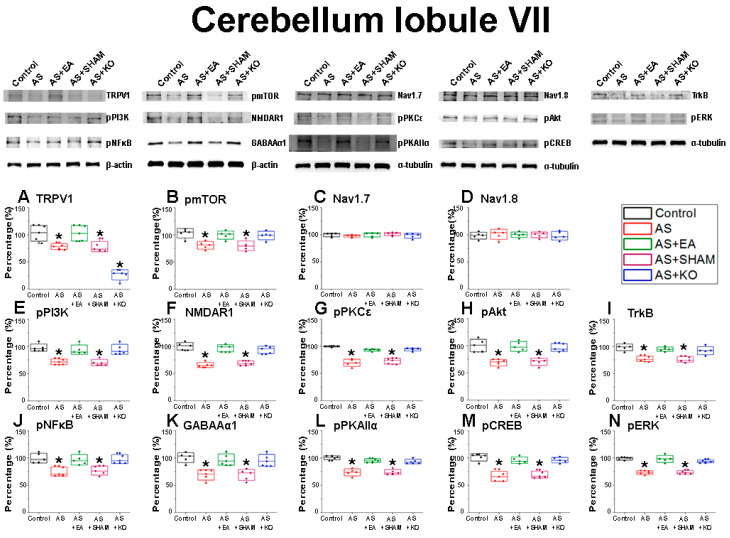
The expression levels of nociceptive receptors and associated molecules in the cerebellum lobule VII. The immunoblotting images depict five lanes of protein in the following order: Control, AS, AS + EA, AS + SHAM, and AS + KO groups. There are significant decreases in protein expression in the AS and AS + SHAM groups of (**A**) TRPV1, (**B**) pmTOR, (**E**) pPI3K, (**F**) NMDAR1, (**G**) pPKCε, (**H**) pAkt, (**I**) TrkB and (**J**) pNFκB (**K**) GABAAα1, (**L**) pPKAIIα, (**M**) pCREB, and (**N**) pERK levels, which were significantly attenuated in the AS + EA and AS + KO groups, depicting no difference when compared to the Control group. Interestingly, the protein expression levels of (**C**) Nav1.7 and (**D**) Nav1.8 displayed states of no significant variances across all 5 groups. Accordingly, the protein density of the AS + KO group revealed a predicted decrease in (**A**) TRPV1 (* *p* < 0.05).

**Figure 4 ijms-22-05028-f004:**
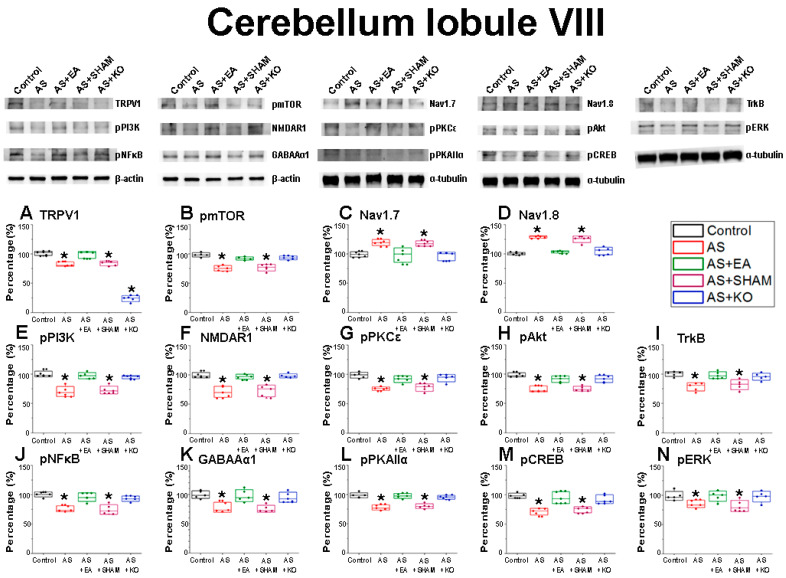
The expression levels of nociceptive receptors and associated molecules in the cerebellum lobule VIII. The immunoblotting images depict five lanes of protein in the following order: Control, AS, AS + EA, AS + SHAM, and AS + KO groups. There are significant decreases in protein expression in the AS and AS + SHAM groups of (**A**) TRPV1, (**B**) pmTOR, (**E**) pPI3K, (**F**) NMDAR1, (**G**) pPKCε, (**H**) pAkt, (**I**) TrkB, (**J**) pNFκB, (**K**) GABAAα1, (**L**) pPKAIIα, (**M**) pCREB, and (**N**) pERK levels, which were significantly augmented in the AS + EA and AS + KO groups. Both the AS + EA and AS + KO groups displayed no difference when compared to the Control group, depicting an observable improvement of CPDC tendencies. However, the protein expression levels of (**C**) Nav1.7 and (**D**) Nav1.8 displayed states of significant increases in the AS and AS + SHAM groups when compared to the Control group. Furthermore, this increase was significantly ameliorated in the AS + EA and AS + KO groups, which retained states of non-significance when similarly compared to the Control group. Accordingly, the protein density of the AS + KO group revealed a predicted decrease in (**A**) TRPV1 (* *p* < 0.05).

**Figure 5 ijms-22-05028-f005:**
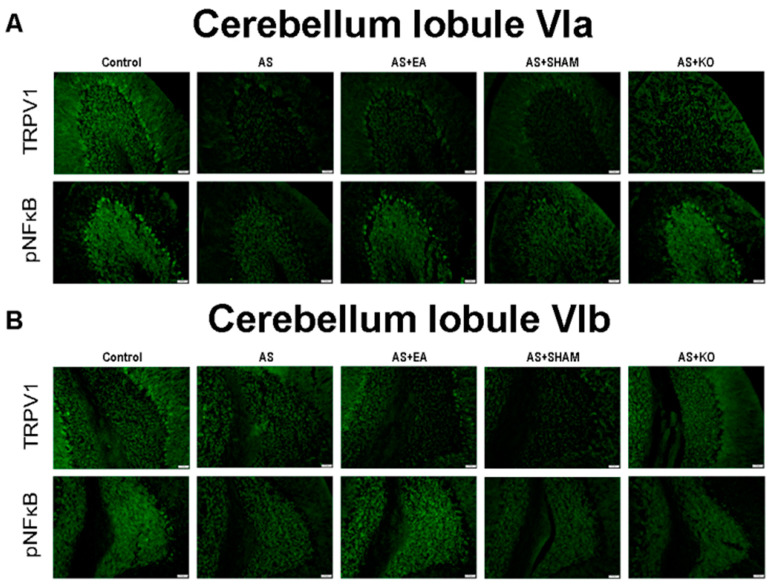
Immunofluorescence staining of TRPV1 and pNFkB protein expression in the cerebellum lobules VIa and VIb. There are 5 subject groups: Control, AS, AS + EA, AS + SHAM, and AS + KO. (**A**) The efficacy of EA treatment involves significant increases in TRPV1 and pNFkB densities in the cerebellum lobule VIa. (**B**) Conversely, no significant variance in pNFkB density was observed in cerebellum lobule VIb among the 5 groups, although TRPV1 maintained an analogous trend of decreased expression in the AS and AS + SHAM group, which was absent in the AS + KO group and increased in the AS + EA group when compared to Control. Scale bar is 50 μm.

**Figure 6 ijms-22-05028-f006:**
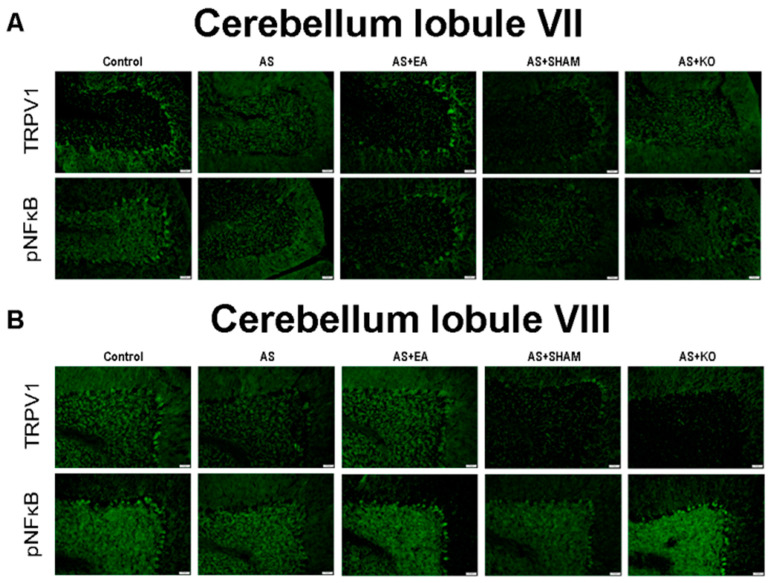
Immunofluorescence staining of TRPV1 and pNFkB protein expression in the cerebellum lobules VII and VIII. There are 5 subject groups: Control, AS, AS + EA, AS + SHAM, and AS + KO. The efficacy of EA treatment involves significant increases in TRPV1 and pNFkB densities in the cerebellum lobules (**A**) VII and (**B**) VIII. Scale bar is 50 μm.

**Figure 7 ijms-22-05028-f007:**
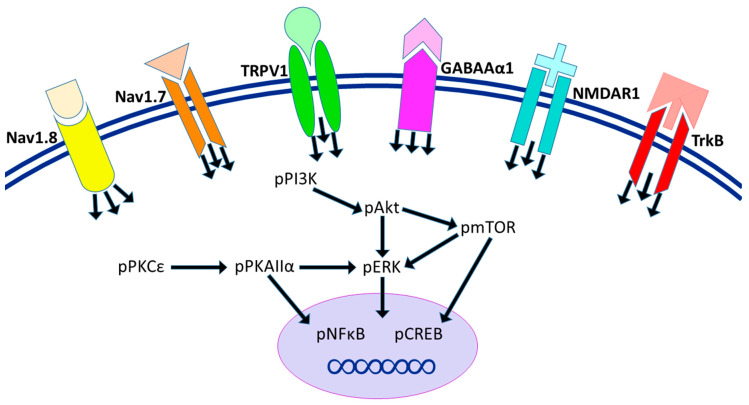
Illustration of CPDC pathways attenuated by EA in the cerebellum. We found that EA at 1 mA, 2 Hz/20 min, and TRPV1 gene deletion can increase the expressions of TRPV1, Nav1.7, Nav1.8, GABAAα1, NMDAR1, and TrkB receptors in the AS-induced CPDC model, as observed in the AS + EA and AS + KO groups. Furthermore, the relevant increases in related responses cause effective increases in the signaling of pPI3K, pAkt, pERK, pmTOR, pPKCε, pPKAIIα, pNFκB, and pCREB under conditions of CPDC, and these serve to present a basic neuromodulatory pathway of CPDC, TRPV1 signaling, and the molecular function of EA.

**Table 1 ijms-22-05028-t001:** Statistical analysis depicting a percentage value of the expression levels of TRPV1-associated signaling pathways and relative proteins within the murine cerebellum lobules VI, VII, and VIII.

**Statistical Analysis Depicting the Expression Levels of TRPV1-Associated Proteins in the 5 Comparative Groups of an Acid-Saline Induced CPDC Model**					
**Cerebellum Lobule VI**					
Protein	Control	AS	AS + EA	AS + SHAM	AS + KO
TRPV1	99.96 ± 2.86	71.75 ± 4.58 *	92.37 ± 2.38	71.22 ± 2.63 *	44.39 ± 8.59 *
pmTOR	99.96 ± 2.16	75.76 ± 1.86 *	96.01 ± 2.13	77.53 ± 1.88 *	96.48 ± 1.96
Nav1.7	100.04 ± 1.65	132.43 ± 3.9 *	105.05 ± 2.64	128.22 ± 3.46 *	106.36 ± 2.62
Nav1.8	100.01 ± 4.63	118.93 ± 4.93 *	102.86 ± 3.93	120.76 ± 2.51 *	105.03 ± 4.35
pPI3K	100.04 ± 1.4	68.04 ± 3 *	96.12 ± 2.95	69.77 ± 2.9 *	94.13 ± 1.58
pNMDAR1	100.04 ± 2.85	76.99 ± 1.95 *	97.51 ± 3.35	76.32 ± 2.59 *	95.11 ± 3.73
pPKCε	100.01 ± 2.52	73.34 ± 4.02 *	96.37 ± 3.59	74.24 ± 4.77 *	92.69 ± 4.04
pAkt	99.97 ± 3.21	71.76 ± 3.78 *	93.72 ± 2.39	68.46 ± 3.21 *	94.95 ± 2.43
TrkB	100.05 ± 3.28	73.17 ± 2.87 *	96.98 ± 4.69	73 ± 3.84 *	96.5 ± 4.84
pNFκB	99.98 ± 0.92	69.43 ± 3.3 *	96.33 ± 3.76	70.31 ± 1.75 *	93.33 ± 2.58
GABA_Aα1_	100.01 ± 2.86	74.93 ± 1.22 *	98.87 ± 2.91	73 ± 2.45 *	98.2 ± 2.29
pPKAIIα	100.03 ± 3.92	69.15 ± 2.02 *	97.7 ± 3.39	72.12 ± 2.63 *	95.42 ± 3.3
pCREB	99.95 ± 5.02	78.85 ± 4.16 *	98.8 ± 5.55	78.28 ± 4.87 *	97.68 ± 6.27
pERK	100.03 ± 2.13	73.25 ± 3.36 *	95.53 ± 2.38	73.1 ± 2.36 *	93.86 ± 3.02
**Cerebellum Lobule VII**					
Protein	Control	AS	AS + EA	AS + SHAM	AS + KO
TRPV1	100.04 ± 5.91	78.04 ± 1.99 *	100.59 ± 5.58	77.51 ± 3.51 *	44.22 ± 1.53 *
pmTOR	99.96 ± 4.15	79.23 ± 2.91 *	96.77 ± 3.81	78.04 ± 3.71 *	95.58 ± 4.23
Nav1.7	99.95 ± 1.25	98.27 ± 0.99	98.46 ± 1.99	101.35 ± 1.12	102.36 ± 1.26
Nav1.8	100.01 ± 2.84	101.91 ± 3.58	101.34 ± 2.21	101.15 ± 2.33	99.59 ± 3.21
pPI3K	100.17 ± 3.32	72.9 ± 2.22 *	96.08 ± 3.77	71.46 ± 2.44 *	96.28 ± 3.82
pNMDAR1	100.04 ± 2.55	67.86 ± 2.67 *	96.75 ± 2.42	70.98 ± 2.08 *	94.87 ± 2.81
pPKCε	99.95 ± 0.43	71.91 ± 3.28 *	93.68 ± 0.96	72.67 ± 1.96 *	93.64 ± 1.16
pAkt	100.06 ± 4.19	69.61 ± 2.38 *	96.84 ± 3.67	69.53 ± 2.52 *	96.59 ± 2.68
TrkB	100.04 ± 2.93	74.01 ± 3.66 *	96.19 ± 2.08	73.79 ± 2.87 *	94.1 ± 3.3
pNFκB	100.02 ± 3.23	76.58 ± 3.3 *	97.37 ± 3.47	76.73 ± 3.05 *	99.53 ± 3.29
GABA_Aα1_	99.97 ± 3.36	69.57 ± 3.54 *	97.55 ± 3.74	69.4 ± 3.76 *	95.07 ± 3.94
pPKAIIα	99.99 ± 1.37	73.36 ± 2.36 *	95.17 ± 1.22	74.34 ± 1.69 *	93.61 ± 1.49
pCREB	100.02 ± 2.8	65.82 ± 3.44 *	96.72 ± 2.22	69.09 ± 2.41 *	95.18 ± 2.03
pERK	100.05 ± 1.66	75.59 ± 2.62 *	99.18 ± 2.44	73.56 ± 1.22 *	92.92 ± 1.49
**Cerebellum Lobule VIII**					
Protein	Control	AS	AS + EA	AS + SHAM	AS + KO
TRPV1	100.03 ± 1.6	81.71 ± 1.69 *	97.31 ± 2.4	82.58 ± 1.88 *	24.38 ± 2.00 *
pmTOR	100.04 ± 2.14	76.94 ± 2.26 *	93.65 ± 1.75	78.75 ± 3.12 *	94.93 ± 2.15
Nav1.7	99.99 ± 1.87	121.07 ± 2.59 *	99.53 ± 4.81	119.73 ± 2.7 *	98.12 ± 3.58
Nav1.8	100.04 ± 0.82	129.31 ± 0.92 *	103.25 ± 0.74	124.2 ± 2.16 *	104.73 ± 2.34
pPI3K	99.97 ± 1.78	70.8 ± 3.19 *	96.86 ± 1.98	73.66 ± 2.55 *	94.66 ± 1.04
pNMDAR1	100.01 ± 1.77	72.77 ± 3.87 *	98.06 ± 1.85	74.93 ± 4.32 *	99.39 ± 1.63
pPKCε	99.96 ± 2.3	74.36 ± 1.67 *	93.14 ± 2.44	78.35 ± 2.51 *	94.14 ± 2.36
pAkt	99.97 ± 1.81	73.82 ± 2.07 *	90.77 ± 2.08	74.55 ± 1.62 *	91.36 ± 2.19
TrkB	99.99 ± 1.52	77.3 ± 2.71 *	98.11 ± 2.14	80.68 ± 3.37 *	95.19 ± 2.25
pNFκB	100.04 ± 1.64	76.11 ± 1.87 *	96.68 ± 2.78	75.38 ± 3.06 *	93.51 ± 1.69
GABA_Aα1_	99.99 ± 2.37	78.08 ± 3.12 *	97.47 ± 3.78	75.95 ± 2.22 *	95.13 ± 3.16
pPKAIIα	100 ± 2.07	77.19 ± 1.79 *	97.39 ± 1.53	79.51 ± 1.62 *	95.58 ± 1.23
pCREB	100.01 ± 2.29	73.12 ± 2.83 *	96.31 ± 3.88	75.73 ± 2.37 *	93.61 ± 2.91
pERK	100.03 ± 2.94	85.53 ± 2.33 *	99.62 ± 3.37	82.59 ± 3.61 *	99.62 ± 3.82

Data are recorded as a percentage value of protein and expressed as Mean ± Standard Error (S.E.M.) with Control standardized to 100%. * Indicates statistical significance when compared to the Control group (* *p* ˂ 0.05, *n* = 6).

## Data Availability

The data presented in this study are available on request from the corresponding author.

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
