# Peer review of "TRPV1 Responses in the Cerebellum Lobules VI, VII, VIII Using Electroacupuncture Treatment for Chronic Pain and Depression Comorbidity in a Murine Model"

_ijms, 2021, doi:10.3390/ijms22095028_

Round 1

Reviewer 1 Report

This study aimed at clarifying the efficacy of EA treatment and characterizing the modification of various molecular substrates in the cerebellum in CPDC disorder. The authors hypothesized that AS injections would cause alterations in the cerebellum due to the CPDC, which would be attenuated through the use of EA at ST36 via actions on TRPV1 and associated downstream molecules. Although the question they tried to address is scientifically important, there are a few things to be addressed before being published in the journal. 1. For immunoblotting, the authors used total protein for the analysis. The authors seemed to claim too much about Figure 7, given that the analysis used total protein. 2. The rationale for focusing on the cerebellum in this work seems to be weak. 3. Some words should be spelled out, such as EA, OFT, when they came firstly in the text.

Author Response

Review comments for Article ijms-1172618: TRPV1 Responses in the Cerebellum Lobules VI, VII, VIII Using Electroacupuncture Treatment for Chronic Pain and Depression Co-morbidity in a Murine Model

Reviewer Reports

Reviewer #1

Open Review 1

This study aimed at clarifying the efficacy of EA treatment and characterizing the modification of various molecular substrates in the cerebellum in CPDC disorder. The authors hypothesized that AS injections would cause alterations in the cerebellum due to the CPDC, which would be attenuated through the use of EA at ST36 via actions on TRPV1 and associated downstream molecules. Although the question they tried to address is scientifically important, there are a few things to be addressed before being published in the journal.

  1. For immunoblotting, the authors used total protein for the analysis. The authors seemed to claim too much about Figure 7, given that the analysis used total protein.

Response: Thank you for your comment. The hypothesis of functions and mechanisms associated with Figure 7 are intended to provide novel evidence as to the possible mechanisms associated with EA. To this end, of the references included support these mechanisms of action to a certain degree. Most notably, the paper published by [1], examines the response mechanisms of EA treatment in inflammatory hyperalgesia, and also defines the possible mechanisms associated therewith in a similarly illustrated figure. Further evidence of this is observed in the publication by [2], whereby the similar tendencies to that of the aforementioned total protein. In chronic pain condition, long term central sensitization indeed induced total protein changes. Furthermore, the support of current literature regarding novel findings underlying the mechanisms associated with EA also serve to support the foundational conclusions of Figure 7.

  1. The rationale for focusing on the cerebellum in this work seems to be weak.

Response: Thank you for your comment. The article has been edited to reflect a better rationale of the focus on the cerebellum. Current references have been included to describe this more relevant detail. These changes can be found on pages 2 and 3, highlighted in yellow and include relevant references. [3][4][5]

  1. Some words should be spelled out, such as EA, OFT, when they came firstly in the text.

Response: Thank you for the comment. The abbreviated words have been edited accordingly in the text and have been spelled out when they first appear. These changes are visible on page 2 and 3, highlighted in yellow.

      1. Inprasit, C. and Y.-W. Lin, TRPV1 Responses in the Cerebellum Lobules V,              VIa and VII Using Electroacupuncture Treatment for Inflammatory                          Hyperalgesia in Murine Model. International Journal of Molecular Sciences,            2020. 21(9): p. 3312.

  1. Lottering, B. and Y.-W. Lin, Functional characterization of nociceptive mechanisms involved in fibromyalgia and electroacupuncture. Brain Research, 2021. 1755: p. 147260.
  2. Gary, Z.Y., et al., Accelerated brain aging in chronic low back pain. Brain Research, 2021. 1755: p. 147263.
  3. Waller, N.C., Regional brain activation in chronic pain: A functional connectivity meta-analysis with healthy controls and chronic pain patients. 2020.
  4. Depping, M.S., et al., Abnormal cerebellar volume in patients with remitted major depression with persistent cognitive deficits. The Cerebellum, 2020. 19(6): p. 762-770.

Reviewer 2 Report

The manuscript “TRPV1 Responses in the Cerebellum Lobules VI, VII, VIII Using Electroacupuncture Treatment for Chronic Pain and Depression Co-morbidity in a Murine Model” by Lottering and Lin examines the effects of EA at ST36 on depression behaviors and the changes of TRPV1 and related molecular pathways in Cerebellum Lobules VI, VII, VIII in CPDC mice model. This work follows up on previous work from Liu’s group showing TRPV1 responses in Inflammatory Hyperalgesia in Murine Model. This paper lacks novel points and requires considerable revision in its current form.

Major considerations:

  1. This paper lacks sufficient evidences to get the conclusion that EA attenuated depression behavior in CPDC. First, the authors only set up AS model, not CPDC model. There is no evidence that all AS mice model accompanied by depression. As mentioned in ‘Materials and Methods’, there are 6 mice in each group. And it is necessary to show how many mice in AS group are accompanied by depression. Second, ‘Materials and Methods’ mentioned that depression behavior tests, OFT and FST, were conducted at Day 0, 7, 14, 21 and 28, but only showed the data on Day 28 in Fig.1. There is obvious individual differences among mice in behavior. Therefore, depressive behavior tests should be conducted before and after AS injection, to get the initial performance and the changes of these tasks of each mouse. Finally, the chronic pain caused by AS injection may affect motion function of mice and lead to inaccurate results of forced swimming and open field tests. Sucrose consumption/preference tests should be added minimally to make a depression behavioral conclusion.
  2. While EA at ST36 in AS model mice can upregulate TRPV1 expression and stimulate related molecular pathways, this paper is simply descriptive with no connection between these changes and the behavior of AS model. This is significant shortcoming. The behavior tasks of EA treatment in AS/KO mice can be conducted to test whether EA could improve behavior of AS/KO mice. If TRPV1 knockout ameliorates the depression behavior, the conclusion ‘EA at ST36 via its action on TRPV1 and related molecular pathways’ can be obtained.

Minor:

  1. In 4.4.2 Depression Behavioral Examination, duration of observation in FST should be mentioned.
  2. There is one more space in last line on page 5.

Round 2

Reviewer 2 Report

The authors have addressed the first major concern in my comments. As for the second, I reserve my opinion.

In response from authors, ‘In summary, information suggests that pathologies can be improved by TRPV1 functional reduction which attenuates the pain mechanisms, in the form of TRPV1 gene deletion or TRPV1 antagonists’.

This statement contradicts the results shown in Fig.2-6 that EA upregulated the expression of TRPV1 in AS+EA group. Why there is no significant difference between AS and AS+KO group in Fig.1, if TRPV1 functional reduction improve the pathologies.

Author Response

Thank you for your comment.

With regards to the specific reduction of TRPV1 in improving pathological responses, this decrease specifically would refer to the attenuation of only pathological nociceptive responses as supported by the aforementioned literature. However, the AS model used in our study depictes states of comorbidity that feature both hyperalgeasia and psychological abnormalities with specificity to depression. To this end, our paper indicates that TRPV1 is an important underlying pathological influencer in the acquisition and expression of CPDC. Moreover, EA ameliorates this condition through its interaction with the TRPV1 channel and related molecules. With respect to the question raised regarding Figure 1 and the lack of effect observed in the TRPV1 KO channel in behavioural examinations, the discussion reflects that the action of TRPV1 is not singular in CPDC, which would suggest other potential targets that could be explored in future studies. Furthermore, we observe that TRPV1 does not necessarily respond to all behaviours. Several receptors/ion channels such as ASIC3 or voltage gated sodium channels were also reported involving in this model.

Round 3

Reviewer 2 Report

Thank you for your response. I hope you get more stronger evidences to support your conclusion. 

Author Response

We revised the conclusion section "In conclusion, we determined significant differences in the pain and depression behaviours in AS-induced CPDC mice. We conclude that AS-induced CPDC directly influences the acquisition and expression of TRPV1, pmTOR, Nav1.7, Nav1.8, pPI3K, NMDAR1, pPKCε, pAkt, TrkB, pNFκB, GABAAα1, pPKAIIα, pCREB and pERK in the cerebellum via either Western blot or Immunofluorescence evidences." (Page 12, highlighted in yellow.)